# Occurrence and Transport of Isothiazolinone-Type Biocides from Commercial Products to Aquatic Environment and Environmental Risk Assessment

**DOI:** 10.3390/ijerph19137777

**Published:** 2022-06-24

**Authors:** Iuliana Paun, Florinela Pirvu, Vasile Ion Iancu, Florentina Laura Chiriac

**Affiliations:** National Research and Development Institute for Industrial Ecology-ECOIND, Drumul Podu Dambovitei 71-73, Sector 6, 060652 Bucharest, Romania; iuliana.paun@incdecoind.ro (I.P.); florinela_pirvu@yahoo.com (F.P.); vasileiancu10@gmail.com (V.I.I.)

**Keywords:** isothiazolinones, biocides, waste water, surface water, risk assessment

## Abstract

This study investigated the occurrence and transport of four isothiazolinone-type biocides from commercial products to wastewater treatment plants (influents, sludges, and effluents) and to natural emissaries (upstream and downstream the wastewater treatment plants) in Romania. All four biocides were determined in personal care and household products, with the highest concentration of 76.4 µg/L OIT (2-octyl-4-isothiazolin-3-one). For environmental samples, three of the four isothiazolinones were determined, CMI (5-chloro-2-methyl-4-isothiazolin-3-one) being the prominent compound for water samples. The maximum concentration of 84.0 µg/L in influent, 122 µg/L upstream, and 144 µg/L downstream the wastewater treatment plants were obtained for CMI. Unlike water samples, in the sewage sludge samples, OIT proved to be the dominant compound, with concentration up to 5.80 µg/g d.w. The extremely high levels of isothiazolinone determined in different WWTPs from Romania may be due to the COVID-19 pandemic situation, during which a much larger amount of cleaning, hygiene, and personal care products was used. The isothiazolinone-type biocides were readily removed from the influents of the five WWTPs, with the mean removal rate up to 67.5%. The mean mass loading value for the targeted biocides based on influent was 20.4 μg/day/1000 people, while the average environmental emissions were 6.93 μg/day/1000 people for effluents. The results obtained for riverine water combine with statistical analysis showed that the anthropogenic activities are the major contamination sources of the surface waters. Preliminary ecological risk evaluation showed that BIT (1,2-benzisothiazol-3(2H)-one), OIT, and CMI could pose a very high risks to different aquatic species living in the receiving aquatic environments.

## 1. Introduction

Biocides are chemicals used to deter, destroy, harm, or control a harmful organism. In Europe, these compounds are regulated by the Biocidal Products Regulation. Biocides are classified into four major categories (disinfectants, pest control, preservatives, or other biocide products) according to European Union Biocidal Products Directive 98/8/EC [1]. Because of their fungicidal, bactericidal, and algacidal properties, isothiazolinone-type biocides are used in many consumers (detergents, shampoos, soaps, or shower gels) and construction (paints, varnishes) products [2,3]. 2-octyl-4-isothiazolin-3-one (OIT), 2-Methyl-4-isothiazolin-3-one (MIT), 1,2-benzisothiazol-3(2H)-one (BIT), and 5-chloro-2-methyl-4-isothiazolin-3-one (CMI) are four of the most used isothiazolinone in Europe as active ingredients, biocides, preservatives, and disinfectants to prevent the growth of bacteria and mold in various types of consumer products [4,5]. Of these, more restrictive regulations have been approved for MIT-only cosmetics and the CMI/MIT mixture in rinsing products such as shampoos or shower gels [6,7]. Due to the fact that MIT and CMI are considered allergens and skin irritants, their use in cosmetics has been limited to the maximum permitted concentrations of 0.0015% for a 3:1 CMI/MIT mixture or 0.01% for MIT [8]. For the same reason, CMI is increasingly being replaced with BIT in cleaning products, adhesives, and paints [9].

After use of these products, isothiazolinone-like compounds are released as organic pollutants directly (while using personal care products and cleaning agents or by leaking isotiazolinone-containing materials such as coatings or paints) or indirectly into aquatic ecosystems through wastewater treatment plants (WWTPs). Due to the side effects they can have on aquatic organisms, it is essential to understand their fate and potential ecological risk in the aquatic environment [10,11,12]. For example, toxicological data reported in the literature show LC_50_ values for a CMI/MIT mixture of 0.003 mg/L for a green algae species, 0.16 mg/L for *Daphnia magna* and 0.19 mg/L for rainbow trout [13]. For BIT, the LC_50_ values for green algae, *Daphnia magna,* and rainbow trout are 0.15 mg/L, 1.35 mg/L, and 1.6 mg/L, respectively, indicating that BIT is less harmful to aquatic organisms than the CMI/MIT mixture [13].

Despite their widespread use, large production volumes, and their negative eco-toxicological properties, very few studies have focused on the study of the presence and fate of isothiazolinones in an aquatic environment.

The objectives of this study were: (1) to determine the content of isothiazolinone-type biocides in various commercial products such as care products and household products; (2) investigation of the occurrence and fate of selected biocides in wastewater treatment plants (influents, sludges, effluents); (3) evaluation of the isothiazolinones removal degree by the WWTPs; (4) evaluation of the mass load and the environmental mass emission of the target compounds; (5) to study the transfer of these compounds to natural emissaries by effluents discharges; and (6) assessment of the potential ecological risk of the targeted biocide in natural receptors. The results obtained in this study can help to understand the fate of these biocides in the environment and can contribute to a better management of these chemical compounds in Romania.

## 2. Materials and Methods

### 2.1. Chemicals and Regents

Analytical grade 2-octyl-4-isothiazolin-3-one (OIT, 99.7%), 2-Methyl-4-isothiazolin-3-one (MIT, ≥94%), 1,2-benzisothiazol-3(2H)-one (BIT, 97%), and 5-chloro-2-methyl-4-isothiazolin-3-one (CMI, 99.9%) were selected as target biocide compounds and were acquired from Sigma Aldrich (Darmstadt, Germany). The physico-chemical properties are given in Appendix A. Formic acid, silica gel, and sodium sulfate was purchased also from Sigma Aldrich (Germany). Methanol and acetonitrile, chromatographic grade, were acquired from Merck Millipore (Darmstadt, Germany). Individual stock solutions were prepared in methanol and kept at 4 °C. The ultra-pure water was produced in-house using a Milli Q water system (Millipore, Molsheim, France). All glassware was rinsed with ultra-pure water prior to use.

### 2.2. Sample Collection

#### 2.2.1. Commercial Products

Various personal care and household products were purchased from local markets. Regarding cosmetics, 4 different shampoos, (SH1-SH4), 4 different hair conditioners (HC1-HC4), 4 digested shower gels (SG1-SG4), and 4 different liquid soaps (LS1-LS4) were chosen. In addition, in the category of cleaning products, 4 different dishwashing detergents (DD1-DD4) were studied. All products were stored in their own containers at room temperature until analysis.

#### 2.2.2. Environmental Sample Collection

Wastewater (influent and effluent) and sewage sludge samples were collected from five urban areas from Romanian. Information regarding the locations and sample codification are given in Appendix A. Different environmental samples were collected in October 2021, including wastewater, sludge, and surface water samples, in triplicate. Influent and effluent were collected as 24-h composite samples, while the dehydrated sludge samples were collected from the drying bed. Surface water samples were collected 50 m upstream and downstream the effluents discharge areas into the natural emissaries, namely Bahlui, Siret, Olt, Ialomita, and Dambovita rivers. All water samples were collected in 1 L glass containers and kept at 4 °C during transport and at −4 °C until analyzed. Sludge samples were collected from 3 to 4 points, mixed, and stored into 400 mL glass containers. Sludge samples were frozen at −4 °C and dried by vacuum cooling at −110 °C using a lyophilizer.

### 2.3. Sample Extraction

#### 2.3.1. Commercial Products Extraction

Approximately 1.0 g of the sample was used for the extraction of personal care and household products, using an adapted method reported in the literature [14]. The samples were weighed into a 50 mL separatory funnel, after which a volume of 10 mL methanol was added and vortex stirred for 10 min. The bottom layer was then collected in 50 mL vials, with this procedure being applied three times. The organic phases were dried using sodium sulfate, and the mixture was centrifuged for 10 min at a speed rate of 5000 rpm. The supernatant was filtered through a 0.45 μm filter and diluted to match the composition of mobile phase and kept at 4 °C until chromatographic analysis (for less than 24 h). Every sample was analyzed in triplicate.

#### 2.3.2. Wastewater and Surface Water Extraction

The extraction of the four targeted biocide from water samples was performed through solid phase extraction (SPE) procedure using an automatic purification system (SPE Dionex Autotrace 280, Thermo Scientific, Waltham, MA, USA), with a capacity of 6 positions for cartridges. Before extraction, all water samples (100 mL for influent and effluent samples and 200 mL for surface water samples) were filtered through 0.45 μm glass-fiber filter membranes, acidified at pH = 2, and extracted using Strata-X polymeric cartridges (500 mg, 6 mL, from Phenomenex), which had been conditioned with 10 mL methanol and 5 mL Milli-Q water (pH = 2). The water samples were load through the cartridges at a flow rate of 20 mL/min and then washed with 5 mL Milli-Q water (pH = 2) at a flow rate of 5 mL/min. Each cartridge was then dried for 10 min. The four biocides were eluted from cartridges using 2 × 10 mL mixture of 0.1% formic acid and acetonitrile (50:50, *v*/*v*) at a flow rate of 1 mL/min. The final extracts were evaporated to dryness in nitrogen atmosphere and re-dissolved in 1 mL ultra-pure water prior to analysis.

#### 2.3.3. Sewage Sludge Extraction

The extraction of sludge samples was performed using ultrasonic assisted extraction (UAE) method. A total of 2 g of dry sludge samples was weighted into a 24 mL glass centrifuge tube and extracted with 2 × 10 mL methanol by ultra-sonication for 20 min at room temperature and centrifuged at 5000 rpm for 10 min. The organic phases were collected, purified on silica gel, filtered, and evaporated to less than 0.2 mL under nitrogen stream and then rebuilt with ultrapure water to a volume of 1 mL. All samples were centrifuged again at 5000 rpm for 10 min to remove any suspended particles.

### 2.4. Analytical Method

Analysis of the targeted biocide compounds was performed through high-performance liquid chromatography using an Agilent 1200 HPLC with a Diode-Array Detection (DAD detector). To separate the 4 target compounds, an Eclipse Plus C18 chromatographic column (150 × 3 mm, 5 μm) from Phenomenex, maintained at a temperature of 30 °C, was used. The injection volume was 10 µL. The target compounds were eluted from the chromatographic column using a mobile phase consisting of 0.1% formic acid (A) and acetonitrile (B) at a flow rate of 1 mL/min. The elution was performed in gradient mode, starting from a mobile phase composition of 90% (B) and maintained for 0.8 min, after which the percentage of organic phase decreased to 50% in 3.2 min, and maintained for another 3 min. After this period, the percentage of acetonitrile was suddenly reduced to 10 and maintained for another 8 min. To rebalance the column, the initial mobile phase composition was returned and was maintained for 5 min before the next injection. The elution order proved to be: MIT (r_t_ = 1.33 min), BIT (r_t_ = 3.81 min), OIT (r_t_ = 7.02 min), and CMI (r_t_ = 7.33 min). The compounds detection was carried out at optimal wavelengths identified after the maximum absorption in UV-VIS spectra: 232 nm for BIT and OIT and 278 nm for MIT and CMI

### 2.5. Data Analysis

#### 2.5.1. Removal of Biocide in WWTPs

Removal capacity of the four biocide compounds in the selected WWTPs was calculated as a percentage value using Equation (1):R% = (C_inf_ − C_efl_)/C_inf_ × 100(1)
where C_inf_ represents the biocide compound concentration values in influent samples, and C_efl_ represents the biocide compound concentration in effluent samples.

#### 2.5.2. Pollution Load and Emission Method

Daily mass loading (DML) and daily mass emission (DME) (mg/day/1000 people) of the 4 biocide compounds were estimated for each WWTPs influents and effluents, using the Equations (2) and (3):DML = (Q × C_inf_)/P(2)
DME = (Q × C_efl_)/P(3)
where Q is to the daily flow rate of WWTP (m^3^/day), C_inf_ and C_efl_ represent the biocide compound concentration values detected in influent and effluent samples as expressed in µg/L, and P is the population served by WWTPs.

#### 2.5.3. Risk Assessment Method

Environmental risk was evaluated for both surface water samples (upstream and downstream the WWTPs) and in effluents. The environmental risk is an important parameter, as it is an easy way to evaluate the risk of the targeted biocide for aquatic organisms. The environmental risk was assessed using the risk quotient (RQ) approach according to the European ecological risk assessment guidelines. The RQ values were calculated by dividing the measured mean and maximum environmental concentration (MEC) to the predicted no-effect concentration (PNEC) of each compound. For the PNEC values, toxicity data based on ecologically relevant toxicity endpoints reported in the literature were considered, and the most sensitive ecotoxicological data obtained were employed (Appendix A, [15,16,17]). PNEC values were calculated using the following equation [18,19]:PNEC = lowest NOEC/AF(4)
RQ = MEC/PNEC(5)
where MEC represents the concentration values determined for each biocide compound in the analyzed samples, and PNEC is the concentration with no effect on aquatic organisms; LC_50_—50% of the lethal concentration, EC_50_—50% concentration with effect on species, and NOEC—concentration with no observed effect on species. An assessment factor of 100 or 10 were used [15,17].

The level of risk was assessed as follows [20,21]:RQ < 0.01, no risk;0.01< RQ < 0.1, low risk;0.1< RQ < 1, medium risk;RQ > 1, high risk.

### 2.6. Quality Assurance and Quality Control (QA/QC)

The analytical method was developed in this study. The detector response proved to be linear in the range between 0.02 and 20 mg/L, with correlation coefficients higher than 0.999 for each compound. Inter-day and intra-day precision, the method recoveries, limit of detection (LOD), and limit of quantification (LOQ) for each biocide are given in Appendix A. To check instrument performance and background pollution, a standard mixture (5 mg/L each), a solvent blank, and a procedural blank were used for each sample batch. None of the four biocide compounds were detected in the blank samples. Due to the complex environmental matrices, the identification of biocides was performed not only using the retention time but also comparing the UV spectra recorded for the compounds identified in the samples with those of the target compounds in the standard solution (Appendix A).

## 3. Results and Discussion

### 3.1. Occurrence of Biocide Compounds in Commercial Products

The occurrence of the four targeted biocide in commercial products was evaluated by analyzing five different personal care and household products. The obtained values are presented in Appendix A. As can be seen in Figure 1, most of the personal care products contained OIT as major compound, 39–89%, while in the household products the dominant isothiazolinone compound was BIT (up to 71%). In shower gel samples, the OIT and CMI concentration values ranged of 21.1–74.0 mg/L and 3.80–30.1 mg/L, respectively. For the other two compounds, BIT and MIT, the determined values were much lower, being up to 7.7 mg/L for BIT and up to 6.7 mg/L for MIT. In liquid soap, the dominant compound was OIT (up to 89%), the concentration levels being situated between 65.4 and 76.4 µg/L. Compared to OIT, MIT (up to 0.6 mg/L), BIT (up to 7.4 mg/L), and CMI (up to 2.4 mg/L) were present in very lower concentration values. For shampoo samples, the percentage between OIT (39%), BIT (35%), and CMI (22%) was more balanced, the concentration values being in the range of 0.6–12.9 mg/L for OIT, 4.0–7.4 mg/L for BIT, and 0.7–6.8 mg/L for CMI, while MIT was detected in concentration values up to 0.6 mg/L. In the hair conditioner samples, the four biocide compounds distribution proved to be very similar with those observed in the shampoo samples, with up to 48% OIT, up to 39% BIT, and up to 11% CMI, and the concentration levels ranged between 10.8–11.7 mg/L for OIT, 9.0–9.4 mg/L for BIT, and <LOQ—4.5 mg/L for CMI. Comparing our results with similar ones reported in the literature, it was observed that they are much lower. Thus, in Switzerland, the concentration of MIT and CMI determined in cosmetic products were in the range of 1.3–133 mg/L for MIT and 4.8 mg/L for CMI, the last one being detected in only one shampoo sample [22]. Higher results were also reported in Belgium for MIT, with concentration values ranging in the domain of 150–188 mg/L for leave-on cosmetics and between < LOQ and 163 mg/L for rinse-off cosmetics. CMI compound was determined only in the rinse-off cosmetics, in the range of < LOQ and 8 mg/L [23]. Comparable values with the one reported in this study were reported in the Netherlands, with up to 1.2 mg/L of MIT and up to 1.7 mg/L CMI determined in shampoo samples [24]. Thus, for a single sample (shower gel—SG3), the maximum concentration of CMI/MIT proved to be much higher than those allowed in Europe and United States, which is 15 ppm in cosmetics [25]. Although BIT and OIT are banned in cosmetics in Europe, U.S., and Asia, significant values were observed in the present study. However, given that these compounds are widely used in various consumer products, and this could be another source of biocides in human exposure, the potential risk to human exposure should not be underestimated.

Unlike the personal care products, in household products, BIT compound proved to be the dominant compound (up to 71%), being determined in concentration between 6.5 and 7.7 mg/L. As for the other three targeted compounds, CMI values were situated between 0.2 and 2.2 mg/L, while for MIT and OIT, the determined concentration values were lower than 1.0 mg/L. In the literature, the concentrations reported for detergent samples were much higher in Switzerland (4.3–10 mg/L for CMI, 3.5–279 mg/L for MIT, 3.8–186 mg/L for BIT, and 7.9 mg/L for OIT) [22] and Spain (1.79–29.0 mg/L MI, 4.19–8.44 mg/L CMI, 116–255 mg/L BIT, and 0.56–1.21 mg/L OIT) [5] and comparable in Belgium except for MIT values, which were also much higher (<1.7 mg/L for MCI, <1.3–181 mg/L for MIT, <0.9–26 mg/L for BIT, and <1.5 mg/L for OIT) [23].

### 3.2. Occurrence of Biocide Compounds in WWTPs

The assessment of the concentration levels for the four biocide compounds was evaluated through analyzing the influent, effluent, and excess sewage sludge samples collected from five municipal WWTPs from Romania. The values determined in both influent and effluent samples are summarized in Figure 2, while the concentrations for individual analytes are given in Appendix A.

#### 3.2.1. Concentration of Biocide in Wastewater Samples

CMI was determined at the highest levels in the influent samples with the maximum concentration of 84.0 µg/L and a mean value of 38.6 µg/L, followed by BIT, with a maximum value of 36.9 µg/L and a mean value of 20.8 µg/L. OIT was determined only in two of the five analyzed influent samples, namely 2.81 µg/L for IF VL and 0.91 µg/L for IF B, while MIT was not detected at all. In effluents, the dominant compound proved to also be CMI, with values ranging between 5.70 and 18.5 µg/L. BIT was also detected in all samples, in concentrations varying between <LOQ and 7.57 µg/L, while MIT and OIT were not determined in any of the analyzed samples.

As can be observed in Figure 2b, only in the IF GL and EF GL was the major compound BIT, and the other ones were dominated by the CMI compound.

Comparing the result obtained in this study with other similar experiments, it was observed that they are higher than those reported in the literature. The closest values to those obtained in the present study were observed in Germany, where the isothiazolinone concentrations values are by an order of magnitude smaller. Thus, in influents from Germany, there were reported values up to 3.2 µg/L for BIT and 0.6 µg/L for CMI, but the presence of MIT was also marked in concentration values up to 0.7 µg/L, while in the effluents, none of the targeted isothiazolinones could be detected [26]. Unlike Romania and Germany, in Thailand and China, ng/L values were reported. In Thailand, up to 164 ng/L BIT and up to 0.96 ng/L OIT were found in influents, and up to 9.29 ng/L BIT and up to 0.49 ng/L OIT were detected in effluents [20]. Values ranging in the domain of 8.1–111 ng/L BIT and 0.18–0.81 ng/L OIT in influents and of <LOD BIT and <LOD—0.19 ng/L OIT in effluents were reported in China [27].

The presence of such high levels of isothiazolinone in wastewater treatment plants can be attributed to the COVID-19 pandemic situation, during which a much larger amount of cleaning (disinfectants, sanitizers, detergents, and other cleaning products) and personal care products (hand soap, body wash, cleaning gel, and shampoo) was used [28,29]. After the use of products containing biocides (personal care products, cosmetics, detergents) as well as their use to remove biofilm deposited on pipes, a higher amount of biocide is discharged with domestic wastewater into the sewer network, reaching its end in WWTPs. Unfortunately, biocides cannot be completely removed by conventional treatment technologies, and effluents also contain a significant amount of biocides.

#### 3.2.2. Removal of Biocide in WWTPs

The removal capacity was calculated both for each compound and for the sum of the targeted compounds for each WWTP. Both WWTPs, namely VL and B, exhibited excellent removal efficiencies for OIT compound, with 100% values. Incomplete removals of BIT and CMI biocides in the five WWTPs were determined, with values ranging from 63.8 to 79.5% for BIT and from 44.6 to 78.0% for CMI (Figure 3). Evaluating the removal efficiencies for the sum of compounds, values higher than 70% were registered only for VL and GL WWTPs, while for the others, the values determined were up to 66.6% for TG, 61.3% for IS, and 58.9 for B.

In Thailand, the removal of biocides in WWTPs were ranging from −29% to 100% for individual compounds, while in China, the mean removal rate of targeted biocides was up to 75% [20,27].

#### 3.2.3. Mass Loading and Mass Emission

The total mass of biocides that enter or exit the WWTPs was calculated based on the measured concentrations in influents and effluents, the average daily flow, and the population served by the WWTP. The values obtained for each biocide compound are given in the Table 1.

The highest values determined for daily mass load was observed for CMI in two of the five WWTPs, namely for VL and IS, where the DML values were 67.0 and 34.1 µg/day/1000 people, respectively, followed by the WWTP TG and B, with values of 19.1 and 15.1 µg/day/1000 people, respectively. For BIT, the DML values were situated in the range 12.9 and 16.4 µg/day/1000 people, with the ascending order of WWTPs according to the DML value being B < VL < TG < GL, while in IS, BIT compounds were not detected.

The environmental emission was evaluated for CMI and BIT based on the biocide compound concentrations values determined in the effluent samples. For CMI, the emission in the receiver emissaries was situated between 2.54 and 15.1 µg/day/1000 people. Values up to 15.1 µg/day/1000 people and 12.7 µg/day/1000 people, respectively, were registered for VL and IS WWTPs, while the lower value was determined for GL WWTP. For BIT, the DME values were lower than 5.26 µg/day/1000 people for all WWTPs.

Similar studies were reported in the literature only by Thailand and China, while in UE, no DML or DME values were found. Thus, in Thailand, the mass loading emission values were estimated to be in the range of 0.79–143 for BIT µg/day/1000 people, values much higher than those obtained in the present study, ranging between undetected and 1.3 µg/day/1000 people OIT. The daily mass emissions were estimated for BIT were very close to the ones obtained in this study, namely up to 5.89 µg/day/1000 people, while for BIT, values up to 1.27 µg/day/1000 were reported [20]. In China, higher values of daily mass loading and mass emission were reported for BIT: up to 109 µg/day/1000 people DML and up to 8.64 µg/day/1000 people for DME. As for OIT, the results were similar with those obtained in the present study: up to 0.62 µg/day/1000 people for DML and up to 0.07 µg/day/1000 people for DME [27]. Although the concentration levels of biocides reported in WWTPs from Thailand and China were significantly lower than those determined in Romania, the DML calculated for BIT was higher, probably due to the parameters used in the calculation of DML, namely the flow rate and the number of people served by the WWTPs in question.

#### 3.2.4. Concentration of Biocide in Sewage Sludge

In sludge samples, three of the four biocide compounds were identified, namely BIT, OIT, and CMI, and the results are given in Appendix A and Figure 4. The major contaminant proved to be OIT (2.04–5.80 µg/g d.w.), mainly due to its water-octanol partition coefficient, log Kow = 2.45, which was much higher than of the other targeted biocide (Figure 4b). This property results in a more pronounced affinity of the OIT compound to adsorb solid particles rather than remaining in the liquid medium. BIT and CMI were detected in greater concentration than LOQ in only one sample. BIT was determined only in the sludge sample collected from VL WWTP and CMI in the sludge sample collected from TG WWTP and in values of 0.53 µg/g d.w. and 2.63 µg/g d.w., respectively. The highest concentration value of biocide compound was observed in the case of TG WWTP, while the lowest one was observed for B WWTP, with the ascending order for the five WWTPs being B < IS < VL < GL < TG, with values of 2.05 < 5.19 < 5.25 < 5.80 < 8.18 µg/g d.w.

Comparing the result obtained for biocide compounds with those reported by other similar studies, it has been observed that Thailand reported higher concentrations of BIT (up to 16.9 µg/g d.w.), while OIT concentration values were lower than the LOQ limit [20].

### 3.3. Occurrence of Biocide Compounds in Surface Water

The presence of biocide compounds in an aquatic environment was evaluated based on the values determined both upstream and downstream from the WWTPs, on the pollution sources, and also on the risk posed to aquatic organisms.

#### 3.3.1. Concentration of Biocide in Riverine Environments

Except for MIT, all three biocide compounds were detected in all sample, both upstream and downstream from the WWTPs, with mean concentrations ranging from <LOQ (BIT) to 144 µg/L (CMI). The concentration levels (mean ± standard derivation, *n* = 3) are summarized in Appendix A. In upstream surface water samples, the highest concentration was found for CMI (112 µg/L), followed by BIT (47.5 µg/L) and OIT (29.1 µg/L). The detection frequencies of three biocides (CMI, BIT, and OIT) were all 100%. The highest value of biocide was observed upstream from the VL WWTP, in the Olt River, while the lowest one was observed upstream from the TG WWTP, in the Ialomita River (Figure 5a).

Downstream from the WWTPs, CMI and BIT were detected at higher concentration than OIT, with their maximum concentrations at 144 µg/L and 81.1 µg/L, respectively, compared to the 43.2 µg/L determined for OIT. Except MIT, all three compounds were detected with 100% frequencies. The highest concentration of biocide compounds was also detected downstream the VL WWTP, in the Olt River, but the lowest value was proven to be detected downstream from the GL WWTP, in the Siret River (Figure 5a).

Regarding the distribution pattern, the surface water samples collected upstream and downstream from the VL and IS WWTP, in the Olt and Bahlui Rivers, were dominated by CMI compound, while in the samples collected upstream and downstream from the GL and TG WWTP, in the Siret and Ialomita Rivers, the major contaminant was BIT (Figure 5b). In Dambovita River, in the samples collected upstream and downstream from the B WWTP, the distribution pattern proved to be approximately equivalent between the three determined biocides. Comparing our results with those reported in similar studies, it was observed that lower levels were reported by both Thailand and China. In Thailand, concentrations in the range of ND and 35.3 µg/L were determined upstream the WWTPs, while values up to 56.2 µg/L were obtained for samples collected downstream the WWTPs [20]. The concentration values reported in China proved to be even lower than those reported in Thailand, being in the range of <LOQ—38.7 and <LOQ—376 ng/L for samples collected upstream and downstream from the WWTPs, respectively [27].

Their widespread uses and the improper disposal of the waste that contains such compounds in their chemical formulation could be the principal sources for the occurrence of biocides in surface waters. Since biocides have shown toxicity on aquatic organisms (Appendix A), their presence at high levels in surface waters may present adverse effects on aquatic biota, the most common being slowing or stopping cellular growing [28]. Moreover, potential health hazards of these organic pollutants for human health by consuming aquatic organisms is not to be neglected, as recent studies suggested a potential cytotoxicity effect on human liver and neuronal cell lines [30,31,32].

Correlations between the targeted biocide concentrations determined in surface waters (upstream and downstream from the WWTPs) and effluents were calculated using Spearman rank correlation coefficients (*p* < 0.05). The relationship between the higher values determined downstream the WWTPs and ones determined in both upstream and effluents samples were calculated. The Spearman rank correlation coefficients are shown in Table 2. Significant positive correlations were observed between the target contaminants determined in the samples collected upstream and downstream the WWTPs (Spearman coefficients in the range of 0.9739–0.9920 and *p*-values lower than 0.05). A single exception was identified for the TG location, where the concentrations determined downstream from the WWTPs were strongly correlated with the ones determined in effluent samples (Spearman coefficients = 0.9995, *p* = 0.021). Thus, in the locations VL, IS, B, and GL, the principal source of contamination provided from the loading of rivers with biocides, even before WWTPs, is probably due to improper storage or disposal of waste containing these compounds. For the TG location, the obtained results indicate an important contribution of the TG effluent discharge into the receiving river regarding the contamination of the natural emissary with the biocides in question.

#### 3.3.2. Environmental Risk Assessment

Based on ecotoxicity information reported in the literature (Appendix A), the most sensitive toxicity value identified for each compound was 0.04 mg/L for BIT and 0.0014 mg/L for OIT [15,17]. According to European TGD, for environmental risk assessment of a chemical is based on the lowest NOEC (eventually E/LC50) available in all literature and requires at least three representative trophic levels (microalgae, crustacean, and fish species) [33]. In this study, environmental risk assessment could be evaluated only for BIT and OIT, as the only toxicological data found in the literature for MIT were reported for algae and bacteria. PNEC values for BIT and OIT were obtained from Environment tier II assessment documents where an AF of 100 was used for BIT, as there are incomplete chronic aquatic toxicity data available to fully characterize the toxic hazard of this chemical, and an AF of 10 was used for OIT, there are sufficient acute and chronic aquatic toxicity data available [15,17]. Results regarding the lowest NOEC values, PNECs and RQs for BIT and OIT are given in Table 3 and Appendix A.

The RQ values determined for the studied biocides using both MEC mean and MEC max values were much greater than 1, suggesting a high ecological risks in the aquatic environment. The potential risks to the aquatic environment posed by both biocides indicate the need to identify sources of origin and to develop appropriate risk management for these to organic pollutants. Although the risk to the environment has not been assessed for MIT due to its poor information on its toxicity to various aquatic organisms in the food chain, additional continuous monitoring is recommended before its ecological risk can be characterized.

Given the fact that the marine ecosystem is the final environmental compartment in which a large part of the organic pollutants presents in surface waters end up, the concerns regarding the high risk determined for the studied compounds are not limited to freshwater. A recent critical review focusing on the occurrence, effects, and environmental risk of antifouling biocides in the marine environment, reported literature data regarding very high RQ values determined for DCOIT for marine organisms worldwide (4,5-Dichloro-2-octylisothiazol-3(2H)-one—a compound from the same targeted family in the study) [34]. As example, the ascending order of RQ values determined for DCOIT in the marine environment in different countries was as follows: 7.5 in Sweden [35] < 15 in Korea [36] < 27.5 in Japan [37] < 123 in Greece [38] < 708 in Denmark [39] < 9250 in Spain [40]. Although the environmental risk assessment posed by the selected isothiazolinone compounds to aquatic organism are quite rare, a similar study performed in China reported also medium-to-high risks posed by BIT along with other biocides, including carbendazim, climbazole, and thiabendazole [41].

Given all these results, we can say that the biocides studied can pose a high risk for the aquatic organisms (freshwater and marine organisms) and, reaching the top of the food chain through bioaccumulation and biomagnification processes, can be implicitly dangerous for human health. Considering the large quantities of the targeted biocides in the rivers, potential negative effects on aquatic ecosystems should not be neglected, and caution is needed to reduce their levels.

## 4. Conclusions

The aim of this study was both to assess the level at which four isothiazolinone-type biocides can be found in various commercial products and their transfer to wastewater treatment plants, which are subsequently discharged into riparian systems. The results showed the detection of the four targeted biocides in all personal care and household products analyzed. It was observed that OIT was the major compound in the cosmetic samples, comprising 39–89%, while BIT was the dominant compound in the household products, representing up to 71%. Three of the four target biocides in this study proved to be widely present both in WWTPs and in the surface water samples. Both in influent and effluent samples, the major contaminant was CMI, with values up to 84 and 18.5 µg/L, respectively, while for sludges, higher values of OIT were observed (2.04–5.80 µg/g d.w.). Despite any expectation, values higher than the one determined in the wastewater samples were quantified in the riverine water, both upstream and downstream from the WWTPs. The dominant biocide compound was CMI (up to 112 µg/L in upstream samples and up to 144 µg/L in downstream samples), but significant values were also determined for both OIT and BIT. Statistical analysis suggested good correlation between the biocide concentration levels detected upstream and downstream from samples, and the WWTPs were proven not to be the major source of surface water contamination but rather the anthropogenic contamination of rivers. The environmental risk assessment draws attention to the very high risk of the three biocides determined in this study. The lack of ecotoxicity data available in the literature suggests the need for special attention to these types of studies due to the potential risks of these compounds to aquatic organisms.

## Figures and Tables

**Figure 1 ijerph-19-07777-f001:**
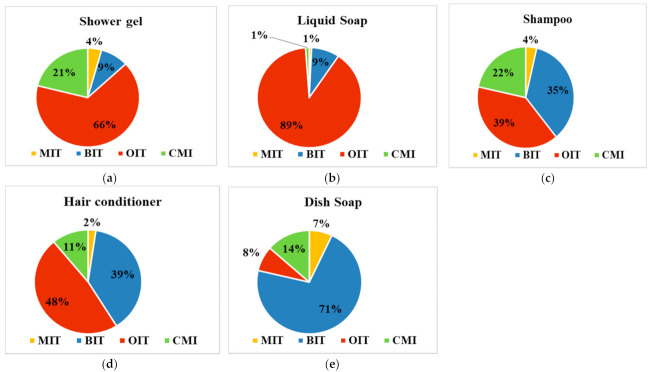
Percentage of biocide compounds in commercial products: (**a**) shower gel, (**b**) liquid soap, (**c**) shampoo, (**d**) hair conditioner, and (**e**) dish soap.

**Figure 2 ijerph-19-07777-f002:**
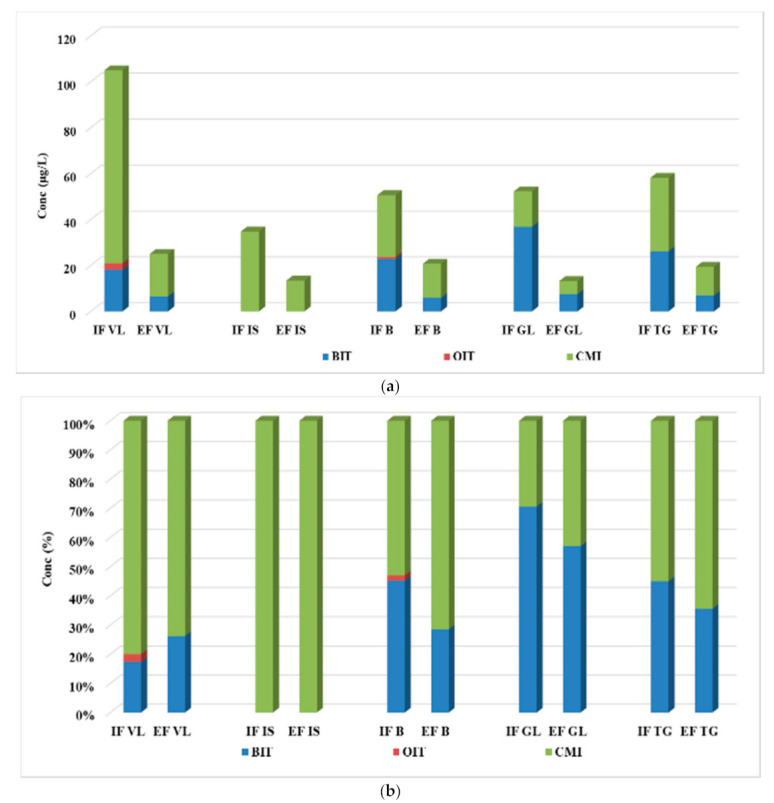
Variation (**a**) and distribution pattern (**b**) of the four-target biocide in wastewater samples.

**Figure 3 ijerph-19-07777-f003:**
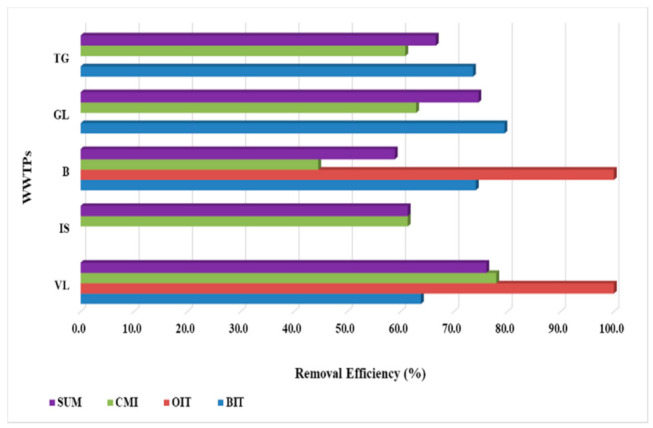
WWTPs removal capacity calculated for both individual compounds and for the sum of them.

**Figure 4 ijerph-19-07777-f004:**
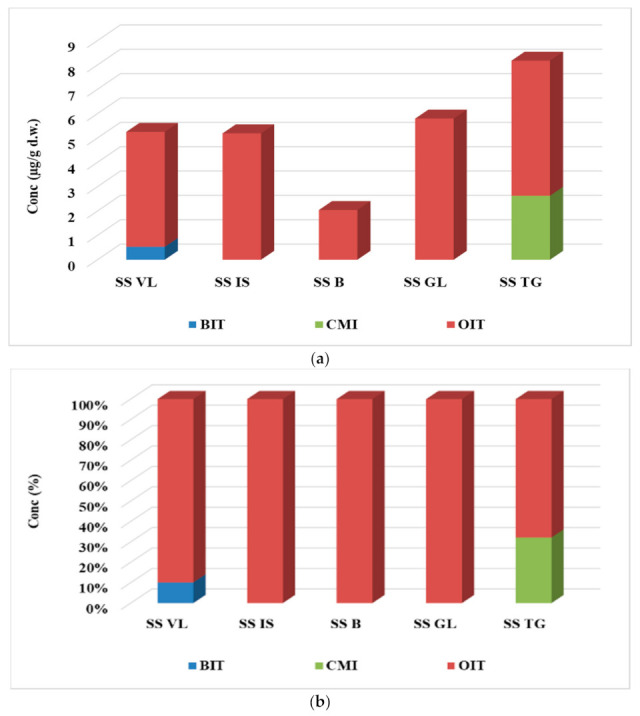
Variation (**a**) and distribution pattern (**b**) of the four target biocides in sludge samples.

**Figure 5 ijerph-19-07777-f005:**
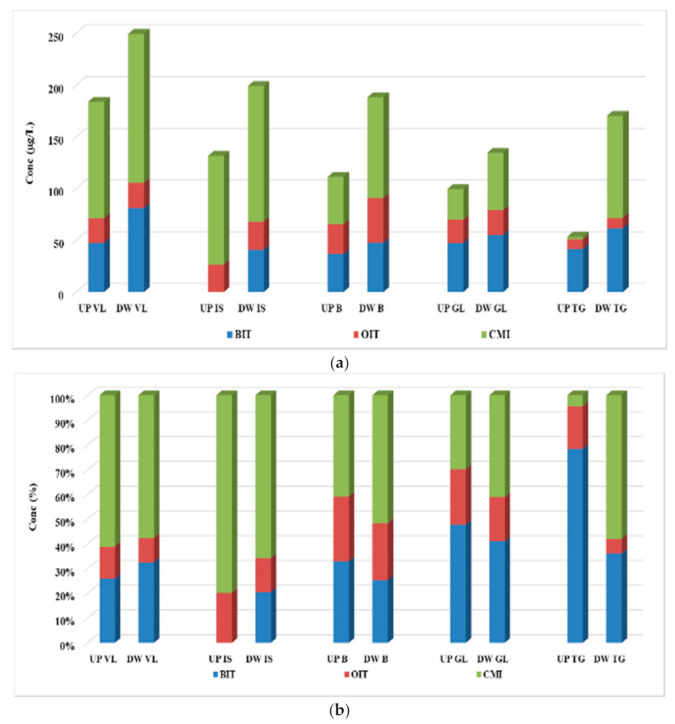
Variation (**a**) and distribution pattern (**b**) of the four target biocides in surface water samples.

**Table 1 ijerph-19-07777-t001:** Daily consumption levels and environmental emission of the 4 biocide compounds (µg/day/1000 people).

WWTPs	DML			DME		
	µg/day/1000 People					
	BIT	OIT	CMI	BIT	OIT	CMI
TG	15.7	-	19.1	4.13	-	7.48
GL	16.4	-	6.86	3.39	-	2.54
B	12.9	-	15.1	3.31	-	8.37
IS	-	-	34.1	-	-	12.7
VL	14.5	2.23	67.0	5.26	-	15.1

**Table 2 ijerph-19-07777-t002:** Spearman correlation coefficients of biocide compounds in the aquatic environment.

	DW VL	DW IS	DW B	DW GL	DW TG
UP VL	0.9906 (*p* = 0.044)				
EF VL	0.9940 (*p* = 0.070)				
UP IS		0.9920 (*p* = 0.023)			
EF IS		0.9327 (*p* = 0.077)			
UP B			0.9895 (*p* = 0.046)		
EF B			0.9447 (*p* = 0.213)		
UP GL				0.9739 (*p* = 0.048)	
EF GL				0.9204 (*p* = 0.155)	
UP TG					−0.0697 (*p* = 0.956)
EF TG					0.9995 (*p* = 0.021)

Green color highlight *p* values lower than 0.05.

**Table 3 ijerph-19-07777-t003:** RQs values derived from BIT and OIT.

Biocides	MEC Mean (µg/L)	MEC Max (µg/L)	Lowest NOEC (mg/L)	AF	PNEC (µg/L)	RQ Based on MEC Mean	RQ Based on MEC Max
BIT	32	81	0.04 ^a^	100	0.4	80	203
OIT	59	144	0.0014 ^b^	10	0.14	421	1029

^a^ Based on algal 72-h NOEC [23]; ^b^ Based on algal 21-d NOEC [25].

## Data Availability

Not applicable.

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
