# Peer review of "Occurrence and Transport of Isothiazolinone-Type Biocides from Commercial Products to Aquatic Environment and Environmental Risk Assessment"

_ijerph, 2022, doi:10.3390/ijerph19137777_

Round 1
Reviewer 1 Report
Dear Authors,
The manuscript is now much improved and reads much better. It can be accepted in the present form, however, I would kindly recommend the following two minor editions:
Lines 54 and 55: rainbow trout is a common name (and not a species name), thus, it must not be italicized (while Daphnia magna must keep the italics).
Line 446: "can pose a high risk for all aquatic organisms (freshwater and marine organisms)" can be replaced by "[...] can pose a high risk for the aquatic organisms [...]".
Best regards
Author Response
Q: Lines 54 and 55: rainbow trout is a common name (and not a species name), thus, it must not be italicized (while Daphnia magna must keep the italics).
A: modified accordingly
Q: Line 446: "can pose a high risk for all aquatic organisms (freshwater and marine organisms)" can be replaced by "[...] can pose a high risk for the aquatic organisms [...]".
A: modified accordingly
Reviewer 2 Report
Authors reported an extraordinarily high concentrations of selected isothiazolinones in river water and wastewater samples than those reported in other countries (Germany, Thailand, and China) by 1-3 orders of magnitude. It is very surprising and may be highly specific in the sampling site of Romania. Chromatograms shown in the Supplementary material are supportive to those high concentrations of isothiazolinones, but are also very unusually clean for water and wastewater analysis.
A few more specific comments are found below.
Line 14-19: The sentence is too long and difficult to understand.
Line 23: than -> that
It requires to explain any potential reasons why extraordinarily high concentrations of isothiazolinones were found in this study in Abstract.
Line 267-281: This part should be re-written. The cited references reported at least 1-3 orders of magnitude lower values for selected isothiazolinones. They are not "similar" as written. In addition, they used mass spectrometry, which is generally considered to be more reliable than uv absorption for the quantification of trace organic contaminants in water samples.
Author Response
Q: Line 14-19: The sentence is too long and difficult to understand.
A: The sentence was modified
Q: Line 23: than -> that
A: modified accordingly
Q: It requires to explain any potential reasons why extraordinarily high concentrations of isothiazolinones were found in this study in Abstract.
A: An explanation has been added
Q: Line 267-281: This part should be re-written. The cited references reported at least 1-3 orders of magnitude lower values for selected isothiazolinones. They are not "similar" as written. In addition, they used mass spectrometry, which is generally considered to be more reliable than uv absorption for the quantification of trace organic contaminants in water samples.
A: The experiments performed in Germany, Thailand and China are similar with the one performed by us. The results are very different, as you mentioned. We re-write the paragraph to be better understandable.
Round 2
Reviewer 2 Report
If authors believe in that the extraordinarily high level of isothiazolinones is due to increased use of personal care products during the Covid-19 pandemic, they should provide appropriate references (e.g., sales data showing the dramatic increase of personal care products) because this is the main result of the paper.
Author Response
two references have been added
This manuscript is a resubmission of an earlier submission. The following is a list of the peer review reports and author responses from that submission.
Round 1
Reviewer 1 Report
Dear Editor and dear Authors,
The paper "Occurrence and transport of isothiazolinone-type biocides from commercial products to aquatic environment and environmental risk assessment" is very interesting, very well written, and provides novel insight on the occurrence and environmental risk of such biocides in Romania. Detected levels in water of such biocides are in the order of mg/L demonstrating the importance of such studies to cover the lack of data and to stress the risk that they pose to the biota and humans.
Despite the overall good quality of the manuscript some comments and suggestions are listed below:
Species names (e.g., line 54 and so on) should be always reported in italics.
Objectives are ambitious for a single manuscript. Nevertheless, the last sentence can be so narrow. The research approach and the outputs of this manuscript will be definitely useful outside Romania.
Material and methods
Line 110. Please, provide units of the centrifugation speed.
Line 112. Please, specify under which conditions samples were kept until their analysis.
Line 116. Please, correct the brand name of the SPE purification system.
Line 124. Please, replace the comma with a dot in the percentage of formic acid.
Line 129. Please, add at the beginning of the sample “A total of [2 g of dry…].”
Lines 148-9. Please, replace all commas with dots in the retention times.
Results and discussion
Lines 165 and 167. Please, replace the minus symbol “-“ with the “–“ which is more adequate when reporting a range.
Line 187. Please, correct the country name to the Netherlands.
Line 208, 391. Please, correct “fourth” by “four”
Lines 225-233. Discussion of the team’s findings would benefit from a true discussion of the data, e.g., for instance, by trying to understand why isothiazolinones levels are so high compared with other studies across the globe.
Lines 238 and 260. This is not a result. Equations and respective supporting text must be placed in the material and methods section, in a subsection of data analysis.
Graphs. The interpretation of the graphs would be easier using the same biocides color across graphs. Additionally, Figure 3 is missing the legend of each color.
Lines 267-278. No discussion was provided. Are these values in line with other EU countries? Are these values acceptable for humans? Can they harm humans?
Lines 324-330. Once again, data is merely compared with literature without going further on the discussion of such important findings. Why are studied biocides levels so higher in Romanian surface waters? Without overlapping section 3.3.2, other questions may be answered, such as, “Can such values harm aquatic biota?”
Lines 331-335 and 350-367. These sentences must be moved to the Material and Methods section. Equation number 4 should be reported as follows: PNECNOEC=lowest NOEC available in the literature/AF
PNECE/LC50=lowest E/LC50 available in the literature/AF
Regarding the risk categories (363-367) please, provide a suitable reference to support your choice.
Lines 368-386. The calculation of the RQ is not correct. According to European TGD there is two ways to derive the ERA of a given chemical: deterministic approach (based on the lowest NOEC (eventually E/LC50) available in all literature, requiring at least ecotoxicological data for one microalgae, one crustacean, and one fish species) and statistical approach (based on species sensitivity distribution curves applicable when a diverse ecotoxicological dataset (>8 groups) was obtained by the authors or it is already available in the literature). Following this rationale, Table 3 makes no sense at all; the RQ must be calculated for the environment and not each species.
Peaking the case of OIT, authors can retrieve the lowest NOEC and the lowest E/LC50 and then divide them by the respective AFs and provide two PNEC values (one based on NOEC and the other on E/LC50); then readers, in particular, regulators would pick the PNEC value that protects better the environment. In the case of CMI, there is a chance that you cannot derive the PNEC value(s) according to the reported data in the literature (unless authors may find data for crustaceans and fish).
On the other hand, authors do not explicitly mention if there are (or not) PNEC values in the literature. If there is, the authors can cite these values and proceed with the risk assessment by dividing the MEC on this study for each biocide by the respective PNEC. Having said this, the entire subsection and respective must be re-written and table 3 reformulated accordingly.
Conclusions
Line 397. Please, explain and contextualize the expression “Despite any expectation” used in this sentence.
Taking this into consideration, and the clear need to improve the discussion and redo the hazard and risk assessment here reported, I only recommended the publication of this manuscript only after a major revision.
Best regards.
Reviewer 2 Report
In this study, authors measured the content of four isothiazolinone biocides (i.e., MIT, BIT, OIT, and CMI) in consumer products as well as in environmental waters (wastewater influents and effluents and river water). Using the measured environmental concentrations and reference toxicity data, they calculated risk quotients for those biocides. Although authors argue that their RQs are very greater than 1, indicating high risks in aquatic environment, results are not very reliable because their concentrations in environmental matrices were determined using HPLC-DAD method after solid-phase extraction (SPE).
It is well-known that those isothiazolinones are highly reactive. They react with thiols in biological materials and undergo ring-cleavage reactions. In the manuscript, no quality control/quality assurance was mentioned. Because there are so many other chemical substances that are extracted by SPE and interfere UV absorption, simply monitoring absorption peaks in DAD does not guarantee the existence of those biocides in wastewater and sludge samples. It is highly likely that authors present false-positive detections. In addition, those isothiazolinone biocides have low extraction recovery. Careful extraction and instrumental analysis are required to obtain their environmental concentration and following risk assessment. Mass spectrometric detections are necessary for those biocides.
It would be okay with using HPLC-DAD method for relatively clean matrix of products.